# The Joining of Alumina to Hastelloy by a TiZrCuNi Filler Metal: Wettability and Interfacial Reactivity

**DOI:** 10.3390/ma16051976

**Published:** 2023-02-28

**Authors:** Andrea Baggio, Fabiana D’Isanto, Fabrizio Valenza, Sofia Gambaro, Valentina Casalegno, Milena Salvo, Federico Smeacetto

**Affiliations:** 1Department of Applied Science and Technology—DISAT, Politecnico di Torino, Corso Duca degli Abruzzi 24, 10129 Torino, Italy; 2National Research Council, Institute of Condensed Matter Chemistry and Technologies for Energy CNR-ICMATE, Via de Marini, 6, 16149 Genoa, Italy

**Keywords:** joining, Al_2_O_3_, Hastelloy, active brazing, thermal expansion, feedthrough, batteries

## Abstract

A systematic microstructural characterization of alumina joined to Hastelloy C22^®^ by means of a commercial active TiZrCuNi alloy, named BTi-5, as a filler metal is reviewed and discussed. The contact angles of the liquid BTi-5 alloy measured at 900°C for the two materials to be joined are 12° and 47° for alumina and Hastelloy C22^®^ after 5 min, respectively, thus demonstrating good wetting and adhesion at 900 °C with very little interfacial reactivity or interdiffusion. The thermomechanical stresses caused by the difference in the coefficient of thermal expansion (CTE) between the Hastelloy C22^®^ superalloy (≈15.3 × 10^−6^ K^−1^) and its alumina counterpart (≈8 × 10^−6^ K^−1^) were the key issues that had to be resolved to avoid failure in this joint. In this work, a circular configuration of the Hastelloy C22^®^/alumina joint was specifically designed to produce a feedthrough for sodium-based liquid metal batteries operating at high temperatures (up to 600 °C). In this configuration, adhesion between the metal and ceramic components was enhanced after cooling by compressive forces created on the joined area due to the difference in CTE between the two materials.

## 1. Introduction

Alumina is a ceramic that is widely used in a variety of industries due to its particular properties such as corrosion resistance, high-temperature strength, electric insulation, and low thermal conductivity. In many cases, alumina is also used in combination with different types of metals [1,2,3,4], which makes the study of metal–ceramic joints particularly interesting for different applications. The development of a reliable metal-to-ceramic bond is particularly challenging because of the difference in the coefficient of thermal expansion (CTE) between the two substrates and the poor wetting of most of the metals on ceramic surfaces. In the past, several solutions have been investigated, such as welding, diffusion bonding, and brazing [1,5,6,7].

Active metal brazing is a technique in which the filler metal contains some elements that react with the ceramic surface at high temperatures, thus promoting wetting on the ceramics [8,9]. Titanium and zirconium, which are used in several alloys, such as AgCuTi [10,11,12], CuTi [13], CuSnTi [14,15], and TiZrCuNi [16], are typical active elements that are effective in metal-to-ceramic joints. Ti-based alloys are widely used, in particular in high-temperature and highly corrosive environments [17,18]. Among others, TiZrCuNi filler metal alloys are considered one of the most popular [19] and have been already successfully used to join ceramics such as zirconia, SiC, and SiC matrix composites to metals [20,21,22]. Moreover, these amorphous alloys exhibit a CTE that is close to that of Al_2_O_3_, and its peculiar ductility [18]—potentially beneficial in mitigating the residual thermal stress after brazing—makes them particularly promising in alumina-to-metal coupling.

In the case of highly corrosive environments—such as in the chemical processing, energy, oil and gas, and pharmaceutical areas—the Hastelloy alloy C family is commonly used [23,24,25,26]. Indeed, the presence of Cr ensures the formation of a passive layer that has a high breakdown potential, while Mo promotes repassivation mechanisms [27,28,29]. Hastelloy C22^®^ (H-C22) is the first alloy of this family that was designed with a high content of Cr in order to enhance corrosion resistance [30]. Therefore, the possibility of coupling corrosion-resistant Ni superalloys with alumina could be particularly interesting for all applications in which high-temperature and corrosion resistances need to be coupled, such as in the energy and aerospace industries [31,32].

The novelty of this work concerns the design and the microstructural characterization of dissimilar joints between alumina and H-C22 for the production of a feedthrough to be used in sodium-based liquid metal batteries operating at high temperatures (up to 600 °C [33,34,35,36]), where hermeticity and corrosion-resistant characteristics are required [34]. The ceramic inner tube, in particular, guarantees the electric insulation of the inner pin electrode from the external case to avoid short circuits, while the metal external part needs to be corrosion-resistant. Indeed, the presence of Na liquid and vapor, as well as liquid Na salts, may negatively affect the feedthrough hermeticity through both the corrosion of the shell materials and the deterioration of the quality of the sealing. For these reasons, the development of the coupling of two corrosion-resistant materials, such as alumina and H-C22, can represent a great opportunity to face two of the main challenges for liquid metal batteries, as reported by Kim et al. in their review [34]: the identification of corrosion-resistant cell components and the design of reliable seals.

To the best of our knowledge, alumina/H-C22 joints have never been tested before. In this work, a commercial active TiZrCuNi alloy, named BTi-5, which was specifically designed for alumina-to-metal joints [37], was selected as the filler metal. First of all, the chemical compatibility of the brazing material with the metal and the ceramic substrates was evaluated with a contact angle (CA) analysis. Despite an excellent wettability, as commonly occurs in most of the metal-to-ceramic joints, the main challenge that has to be addressed is related to the thermo-mechanical stress generated by the difference in CTE between the superalloy (≈15.3 × 10^−6^ K^−1^) and the ceramic counterpart (≈8 × 10^−6^ K^−1^). In the designed circular joined structure, during the cooling process, the external metal ring is expected to apply compressive stress onto the filler metal (CTE ≈ 8.5–8.8 × 10^−6^ K^−1^) and the inner ceramic ring, thereby creating a more intimate contact between the surfaces of the substrates and the filler metal. In order to separately evaluate the thermo-chemical compatibility between the filler metal and the two substrates, some planar joints between H-C22/H-C22 and Al_2_O_3_/Al_2_O_3_ were also studied. The joined samples, as well as the one obtained from the CA analysis, were characterized by means of SEM, FE-SEM, EDS analysis, and the Vickers indentation method.

## 2. Materials and Methods

We used 99.7% pure alumina, provided by VS & S s.r.l (Mesero, MI, Italy) as a substrate for both configurations. The alumina for the planar configuration was in the form of a small 10 × 15 × 3 mm block, while a ring with a 3 mm inner diameter and 7 mm outer diameter was chosen for the circular configuration (Figure 1). Hastelloy C22^®^, whose composition is reported in Table 1 and whose melting temperature is commonly ranged between 1350 °C and 1400 °C, was purchased from Oric Italiana S.r.l. (Castel San Giovanni, PC, Italy) in two different shapes: a 12 mm diameter rod and a 300 × 200 × 3 mm plate. These materials were then cut and drilled into small blocks with the same dimensions as the alumina samples (10 × 15 × 3 mm), and into small tubes with an inner diameter of 7.2 mm and a thickness of 3 mm.

The brazing alloy selected for this study is a commercial product named BTi-5, which was specifically designed for alumina-to-metal joints and is commercialized by Titanium Brazing Inc. (Columbus, OH, USA). This is an amorphous Ti alloy in the form of a 70 µm thick foil, which has its nominal composition reported in Table 2, and solidus and liquidus temperatures of 845 °C and 863 °C, respectively.

The surface features of the alumina and H-C22 plates were quantitatively characterized on 877 × 660 µm areas by means of the confocal technique, using a 3D noncontact profilometer (Sensofar S-neox, Terrassa, Barcelona, Spain), working with a vertical resolution of 1.5 nm (the green light was selected). Quantitative measurements of the average surface roughness were performed according to ISO 25178 [38], using the software embedded in the system (SensoSCAN, Sensofar Metrology, Terrassa, Spain) in order to extract the surface roughness.

Wetting tests of the liquid BTi-5 alloy were performed on the alumina and H-C22 with the sessile drop method, in a tubular alumina furnace equipped with an optical line and a CCD camera [39]. Prior to the experiments, small drops of the BTi-5 alloy, weighing about 0.4 g, corresponding to a volume of ~0.1 cm^3^, were premelted in an arc melting device. The wettability tests were performed under a vacuum (<5 × 10^−4^ Pa) at 900 °C; the alloy/substrate couples were introduced into the preheated furnace by means of an externally operated push rod. After melting, the drops were kept at the testing temperature for 5 min and then quickly brought to room temperature in ~30 s. The mass loss due to evaporation was measured and found to be negligible.

Three types of planar joined samples were produced: two specimens were made with the same substrate by brazing two alumina plates and two H-C22 plates, separately, while the other sample, consisting of the two dissimilar materials, was made using the alumina–C22^®^ plates. The first two samples were separately used to evaluate the thermo-mechanical compatibility of the filler metal with the two substrates. The planar samples were prepared by placing two BTi-5 foils (70 µm thick each) between the two substrates and introducing an external load of 2.28 × 10^3^ Pa. Furthermore, the circular configuration was obtained by rolling three turns of alloy foil around the inner alumina ring and fitting it inside the outer metal ring (Figure 1). All the samples were brazed at 900 °C for 12 min in a high-vacuum furnace (XVAC, Xerion Berlin Laboratories GmbH, Berlin, Germany) at a heating rate of 350 °C/min. A dwell was maintained for 20 min at 750 °C to equalize the temperature inside the chamber (during the heating) and release thermal stresses (during the cooling). A vacuum grade of at least 5.5 × 10^−3^ Pa was maintained inside the furnace throughout the entire process.

The morphology of the joined samples was characterized by analyzing their cross-sections with a scanning electron microscope (SEM) JCM-6000 plus (Joel, Peabody, MA, USA) and a field-emission scanning electron microscope (FE-SEM, Merlin electron microscope, ZEISS, Oberkochen, Germany), equipped with an energy-dispersive spectrometer (EDS) (EDS, Zeiss Supra TM 40, Oberkochen, Germany), to analyze the composition of the different phases that had formed during the thermal treatment. The cross-section of the joined samples was previously polished using SiC papers (grit size 120–4000), and coated with Pt to obtain a conductive surface.

The Vickers indentation test was performed by a Remet HX 1000 microdurometer (Remet, Casalecchio di Reno, BO, Italy) on the joining interfaces, in three different spots for each interface. The applied load of 1 kg was held for 15 s.

## 3. Results and Discussion

The preliminary surface characterization of the two planar substrates produced the two images reported in Figure 2. The surface roughness was measured as 4.9 and 1.4 μm for the H-C22 and alumina surfaces, respectively. These values are quite high and, as no specific surface treatment was conducted before the wetting and joining tests, both materials appeared to be somewhat rough.

Figure 3 shows the evolution of the contact angles (θ) vs. time the BTi-5 drop was on the solid substrates. Both systems exhibited an evolution of the contact angle over time, which led to good wetting. The final contact angles, after 5 min of liquid–solid contact at 900 °C, were 12° and 47° for the alumina and H-C22, respectively.

Figure 4 shows a cross-sectioned BTi-5/Al_2_O_3_ sample after the wettability test at 900 °C for 5 min: the bulk of the solidified alloy presented a eutectic-like structure (compositions in Table 3), in accordance with the literature that describes quaternary Ti-Zr-Cu-Ni alloys as eutectic alloys, which, due to the mutual solubility in the Ti-Zr and Cu-Ni systems, are constituted by phases of the (Ti, Zr)_x_(Cu, Ni)_y_ type [17,18]. The SEM instrument did not reveal the typical reactively formed phases at the active braze/ceramic interface [37,40].

The dissolution of the H-C22 substrate by the liquid BTi-5 alloy was very minimal: as shown in Figure 5, the zone of mutual solubility at the interface was limited to a thick strip of about 3 µm enriched with Cr and Mo. Apart from this thin layer, no extensive mutual interdiffusion occurred. Again, the bulk of the solidified drop formed as a eutectic-like structure.

The spreading in both nonreactive liquid metal/ceramic and liquid metal/metal systems usually occurs in less than one second; however, in our system presented here, equilibrium was achieved in about 200 and 50 s for alumina and H-C22, respectively (Figure 3). The viscosity values of eutectic glass-forming alloys, such as BTi-5 [41], are higher (η~10^1^ Pa·s) than those of liquid metals and alloys (η~10^−4^–10^−3^ Pa·s) [42,43], so that the spreading is controlled by viscous forces, which result in longer spreading times. Moreover, no polishing or cleaning of the surface was performed, and the advancement of the triple line was therefore hindered by surface asperities [44] and, in the case of H-C22, by the oxide layer.

The as-joined samples were then produced using the previously described procedure (900 °C for 12 min in a high vacuum furnace and 750 °C for 20 min). Some pictures of the joined specimens are reported in Figure 6.

Some criticality arose during the manufacturing of the joints. In particular, during the sealing process, the excellent wettability of the brazing alloy on both substrates caused an overflow of the material from the joint area (which is particularly evident in Figure 6c). Thus, the expected thickness of the joined area was almost halved, as evident from the SEM micrographs of the flat samples reported in Figure 7 and Figure 8. In the circular configuration case, it was observed that when solidification occurred, the brazing foils located close to the inner region significantly shrank. For this reason, it was necessary to use an excess of material to fill the whole circumference of the joining area.

The promising results obtained from the wettability evaluation were confirmed by an observation of the cross-sectioned joined samples. Indeed, both the H-C22/BTi-5 and alumina/BTi-5 interfaces exhibited good adhesion in both the flat and circular configurations, thereby resulting in continuous and homogeneous interfaces. A more complex microstructure was observed at the H-C22/BTi-5 interface due to the longer dwelling time of the joining samples at high temperatures (12 min at 900 °C and 20 min at 750 °C), than the interfaces obtained after the wetting tests, and some characteristic phases formed. At least five areas with different compositions were recognizable in the filler metal from the SEM images (Figure 7) and EDS analysis (Table 4) of the H-C22/BTi-5/H-C22 planar configuration. A continuous ~2 μm thick reaction layer, rich in Ti (~38 at %) and Ni (~42 at %), formed at the H-C22/BTi-5 interface (point 5 in Figure 7b). An area of interdiffusion could also be observed on the H-C22 side, close to the filler metal (area 6 and point 7 in Figure 7b), with a consequent migration of elements from the brazing alloy to H-C22 (Cu, Ti, and Zr), while other elements of the substrate moved to the joining interface (Ni and Cr).

A similar analysis was also conducted on the planar Al_2_O_3_/BTi-5/Al_2_O_3_ joints. Two different regions could be distinguished in the brazing alloy from the micrographs collected in Figure 8: one concentrated mostly at the alumina/alloy interface (area 2 in Figure 8b) and another composed of heavier elements (point 1). As previously discussed for the H-C22/BTi-5 interface, the main difference between the microstructure observed for the sessile drop samples (Figure 4) and that of the joined ones was due to the different thermal treatments. The EDS analysis reported in Table 5 showed a high content of oxygen dissolved into the filler alloy region close to the Al_2_O_3_/braze interface (points 1 and 2 in Figure 8b). Both Ti and Zr are able to dissolve a large quantity of O in the solid phase [40,45,46,47,48]. In previous work, Shapiro observed the formation of a double layer of complex oxides at the interface between alumina and BTi-5 in a Ti-alloy/ceramic joint [37]. The different brazing conditions (lower temperature and applied pressure) used in the present study may be sufficient to justify the impossibility of detecting a similar interfacial product by means of SEM analysis. Nevertheless, the compatibility of the brazing alloy with alumina was not compromised, and excellent alumina-to-alumina joints were obtained.

Concerning dissimilar H-C22/Al_2_O_3_ joints, the flat configuration always resulted in detached samples, while the circular configuration was the only one successfully obtained. Indeed, in the circular design, the external part of the joint (H-C22) showed the highest CTE, which thus acted as a compressive force on the inner ceramic part during cooling and promoted better adhesion between the substrates and the brazing alloy. Vickers indentation tests were performed at both metal/braze and braze/alumina interfaces. The images of the indentation areas are reported in Figure 9. The absence of cracks in the brazing alloy close to the H-C22/braze interface (Figure 9a) shows the presence of a ductile region, which can favor a partial relaxation of residual stresses generated during the sealing process [18]. Figure 9b shows the formation of a crack, which propagated perpendicularly to the braze/alumina interface, while no cracks were observed in the parallel direction or in the ceramic. The results of the Vickers indentation test at the braze/alumina interface showed that the diffusion of oxygen from the ceramic into the braze leads to (i) the formation of brittle phases and (ii) the presence of tensile residual stresses on the metal side, with a consequent state of compression on the ceramic [49]. Furthermore, Figure 9b, corresponding to an indentation at the braze/alumina interface, does not show the propagation of any crack along the interface, thus demonstrating the good adhesion between the Bti-5 and alumina.

The morphology of the circular H-C22/Al_2_O_3_ joints, as characterized by FE-SEM, is reported in Figure 10 and Figure 11. Analogous to what was observed for the planar joints, the circular ones showed continuous and defect-free interfaces on both the metal and ceramic sides. The phases observed at the two different interfaces were morphologically similar to the ones observed at the corresponding interfaces of the flat configuration samples. The compositional maps obtained from the EDS analysis on the BTi-5/H-C22 interface (Figure 10) clearly confirmed the presence of an interdiffusion layer on the H-C22 side to a depth of about 10 µm.

The EDS maps reported in Figure 11 evidence the formation of an area rich in Ti and Ni, which is particularly concentrated close to the interface with Al_2_O_3_.

## 4. Summary

The liquid BTi-5 filler metal from the TiZrCuNi alloy family was demonstrated to have excellent chemical compatibility with both alumina and Ni-superalloy Hastelloy C22^®^, which exhibited contact angles of 12° and 47°, respectively. The interdiffusion of some elements between the substrates and the brazing alloy, even though limited, promoted excellent adhesion in the Al_2_O_3_/Al_2_O_3_ and H-C22/H-C22 joints and showed homogeneous and crack-free joining areas after a thermal treatment conducted at 900 °C as the maximum temperature. The significant difference in CTE between the metal and ceramic substrates made it difficult to achieve a reliable joint between the planar alumina and the H-C22 substrates. However, in this study, it was possible to overcome this issue, and a sound joint was obtained between such dissimilar materials in a circular configuration, ideal for feedthroughs to be used in the harsh environment found in sodium-based liquid metal batteries (high-temperature and corrosive environment). It was also demonstrated that the compressive forces applied by the outer metal ring onto the joining area favor adhesion, thus making it possible to achieve good joints between ceramic and metal substrate with different CTEs up to 7 × 10^−6^ K^−1^.

## Figures and Tables

**Figure 1 materials-16-01976-f001:**
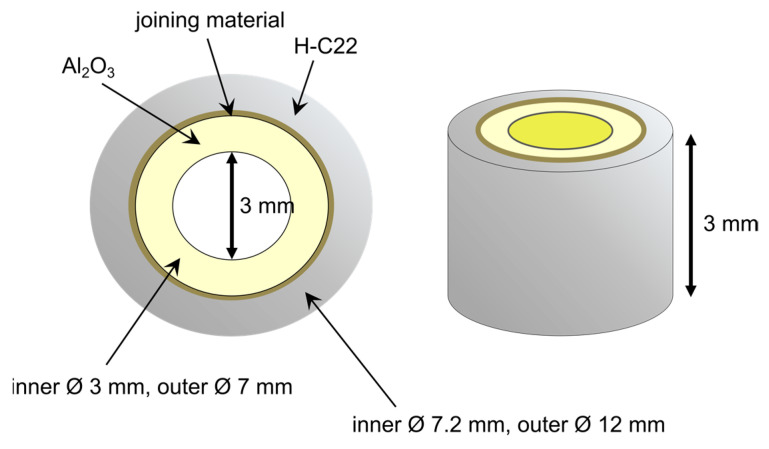
Schematic of the joining configuration. All the sample dimensions are reported.

**Figure 2 materials-16-01976-f002:**
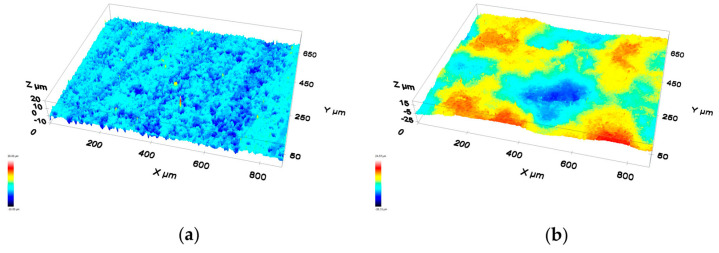
Three–dimensional noncontact profilometry images of the two planar substrates: (**a**) H-C22 and (**b**) alumina.

**Figure 3 materials-16-01976-f003:**
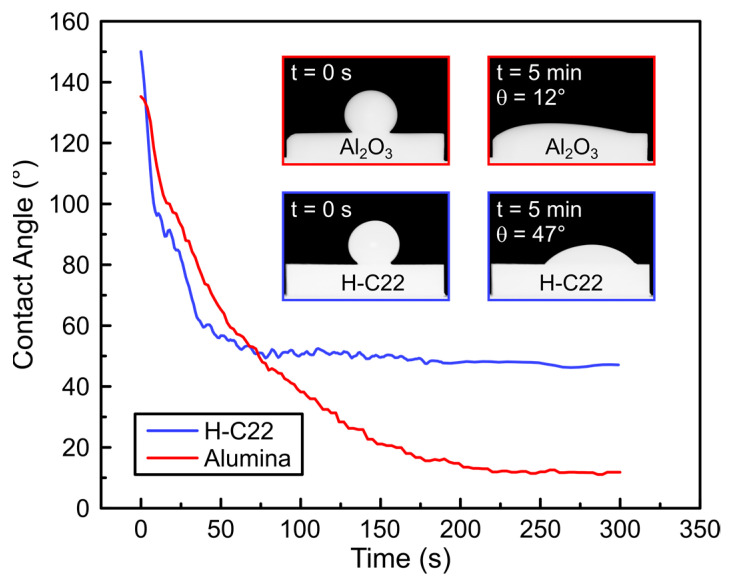
Values of the contact angle of BTi-5 on Hastelloy C22^®^ (blue line) and Al_2_O_3_ (gray line) planar substrates with respect to the time of the thermal treatment carried out at 900 °C in a vacuum atmosphere of at least 5 × 10^−4^ Pa.

**Figure 4 materials-16-01976-f004:**
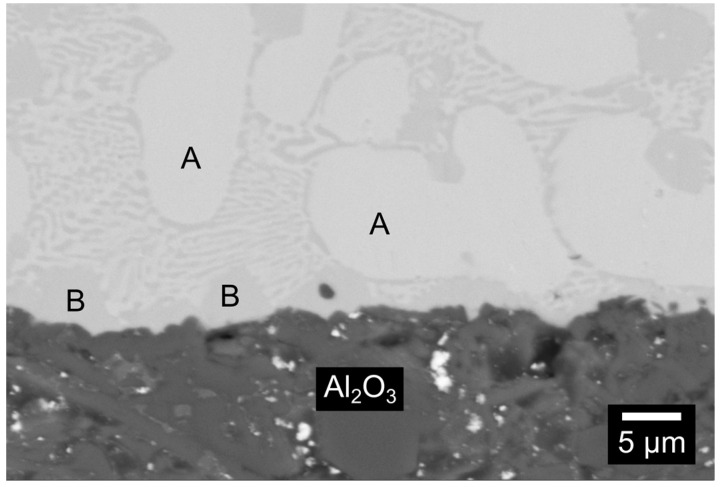
SEM images of the cross-sectioned alumina/BTi-5 interface after the wettability test at 900 °C for 5 min.

**Figure 5 materials-16-01976-f005:**
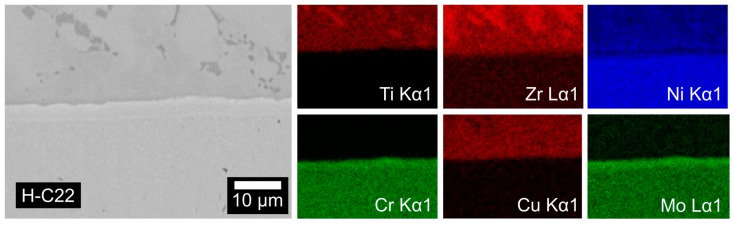
SEM image of the Hastelloy C22^®^/BTi-5 interface and the related element maps after the wettability test at 900 °C for 5 min.

**Figure 6 materials-16-01976-f006:**
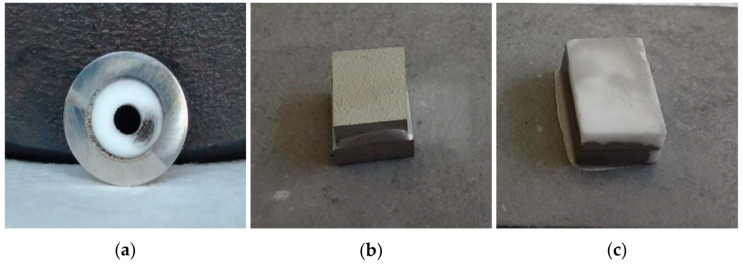
Macrographs of the as-joined samples produced at 900 °C for 12 min in a high-vacuum furnace and 750 °C for 20 min: (**a**) circular configuration, (**b**) H-C22/H-C22, and (**c**) alumina/alumina planar configurations.

**Figure 7 materials-16-01976-f007:**
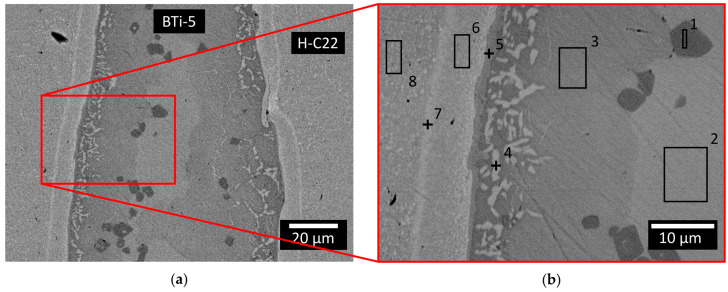
(**a**) SEM-BED images of the cross-section of the planar joining between the two blocks of Hastelloy C22^®^ produced at 900 °C for 12 min in a high vacuum furnace and 750 °C for 20 min, with (**b**) an enlargement of the joining interface. An EDS was performed in this area, and the analyzed points and areas are shown in the picture.

**Figure 8 materials-16-01976-f008:**
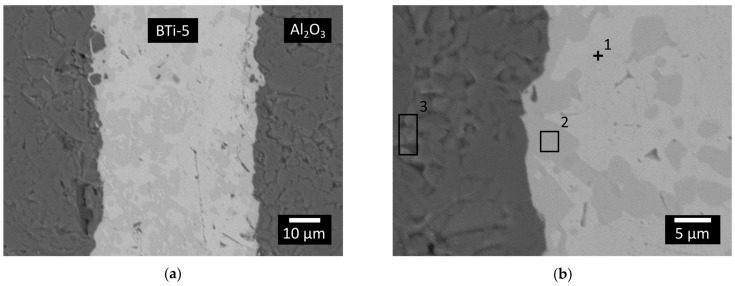
(**a**) SEM-BED images of a cross-sectioned planar joint between the two blocks of Al_2_O_3_ produced at 900 °C for 12 min in a high vacuum furnace and 750 °C for 20 min and (**b**) its related zoomed area. An EDS was performed in this area and the analyzed points and areas are shown in the picture.

**Figure 9 materials-16-01976-f009:**
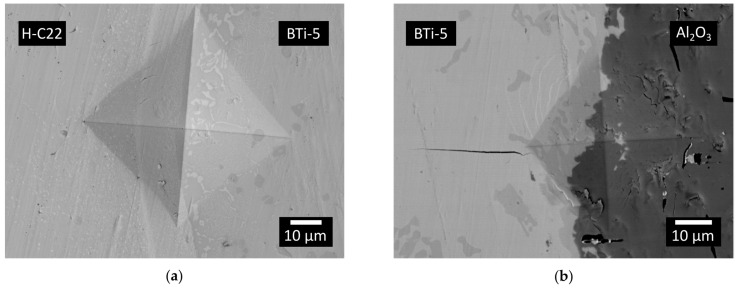
SEM-BED images representative of the Vickers indentation test performed at (**a**) H-C22/BTi-5 and (**b**) BTi-5/alumina interfaces.

**Figure 10 materials-16-01976-f010:**
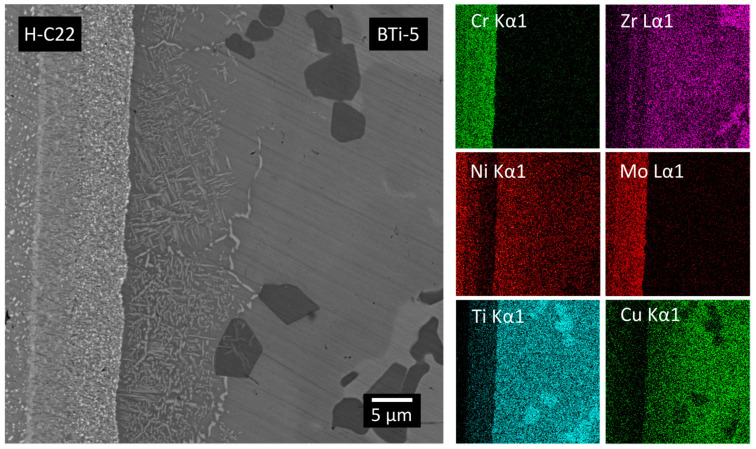
FE-SEM BSE images of the interface of Hastelloy C22^®^ and the filler metal in a circular H-C22/Al_2_O_3_ joined sample produced at 900 °C for 12 min in a high vacuum furnace and 750 °C for 20 min. The EDS maps of the main elements of the two materials.

**Figure 11 materials-16-01976-f011:**
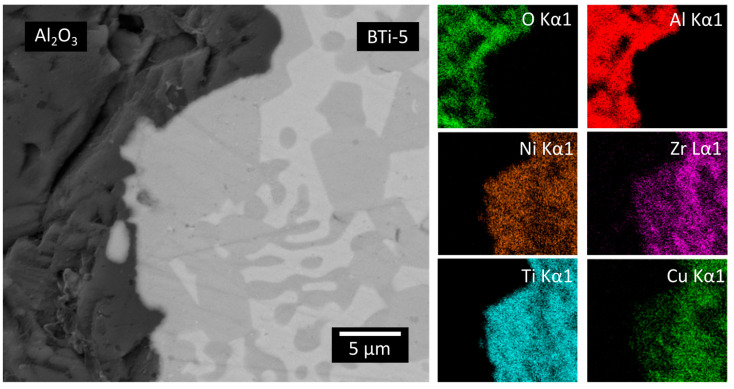
FE-SEM BSE image of the interface of Al_2_O_3_ and the filler metal in the circular joining configuration. EDS maps of the main elements of the two materials.

**Table 1 materials-16-01976-t001:** Range of chemical composition (wt %) of the Hastelloy C22^®^ samples.

(wt %)	Ni	Cr	Mo	Fe	W	Mn	S, Co, P, V, C, Si
H-C22	57.0–58.7	21.3–21.8	13.1–13.6	2.8–5.2	2.6–2.9	0.0–0.2	<0.22

**Table 2 materials-16-01976-t002:** Chemical composition (wt %) of the active brazing alloy BTi-5.

(wt %)	Ti	Zr	Ni	Cu	Hf
BTi-5 foil	39.62	20.20	20.00	19.80	0.38

**Table 3 materials-16-01976-t003:** EDS analysis (at %) of the selected area of the alumina/BTi-5 sample (Figure 4).

Analyzed Area	Ti (at %)	Ni (at %)	Cu (at %)	Zr (at %)
A	44.7	20.7	18.4	16.2
B	61.2	25.0	7.8	6.0

**Table 4 materials-16-01976-t004:** Composition (at %) measured by means of the EDS analysis of the points and areas indicated in Figure 7b.

	Ti (at %)	Cr (at %)	Fe (at %)	Ni (at %)	Cu (at %)	Zr (at %)	Mo (at %)	V + Mn + Co + Hf + W + Si (at %)
1	45.8	-	0.1	22.0	17.6	12.9	0.4	1.2
2	39.9	0.2	0.1	19.0	21.8	16.4	-	2.6
3	37.4	0.1	-	31.9	18.3	9.9	0.5	1.9
4	38.0	0.3	1.4	39.0	11.0	8.5	0.2	1.6
5	37.7	0.4	1.5	42.2	9.6	7.1	-	1.5
6	26.3	10.1	5.4	33.0	11.3	6.5	3.6	3.8
7	14.3	30.1	3.2	34.0	1.9	3.3	11.3	1.9
8	0.4	24.6	5.5	56.7	0.8	1.9	6.3	3.8

**Table 5 materials-16-01976-t005:** Composition (at %) measured by means of EDS analysis of the points and areas indicated in Figure 8b.

	O (at %)	Al (at %)	Ti (at %)	Ni (at %)	Cu (at %)	Zr (at %)
1	13.3	0.9	37.4	17.8	17.5	13.2
2	21.0	0.5	41.6	18.8	11.0	7.1
3	57.7	41.1	-	-	0.1	1.1

## Data Availability

All data are included in the manuscript.

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
