# Peer review of "The Joining of Alumina to Hastelloy by a TiZrCuNi Filler Metal: Wettability and Interfacial Reactivity"

_materials, 2023, doi:10.3390/ma16051976_

Round 1

Reviewer 1 Report

Journal: Materials (ISSN 1996-1944)

Manuscript ID: materials- 2199617

Type: Article

Title: The joining of alumina to Hastelloy by a TiZrCuNi filler metal: wettability and interfacial reactivity.

Authors: Andrea Baggio * , Fabiana D'Isanto , Fabrizio Valenza , Sofia Gambaro , Valentina Casalegno , Milena Salvo , Federico Smeacetto.

a)           Introduction: Write the objective of the present work carefully.

b)          In Table 1 why all element have two values?

c)           For references, choose recent refs. Please, refer to these refs. are very useful for the different measurement characterization

DOI: https://doi.org/10.1088/1742-6596/1795/1/012059

DOI: https://doi.org/10.1016/j.mseb.2021.115191

Best Regards

Reviewer 2 Report

1. The effect of surface roughness on the strength of the joints must be included.

2. The tensile/ bond strength of the joints must be evaluated.

3. The hardness and fracture toughness distribution at the interface / cross-sections of the joints must be included for discussion.

Reviewer 3 Report

Dear Authors,

I have found your paper very interesting for me. I think that it will valuable to other scientists.  I found some points that I would like to clarify with you. 

1. Captions in table 1 and 2 are the same. I think that in table 2 captions should be different.

2. Line 164 . In caption is information about figure 4a. There is no figure 4a in this paper. Please verify it.

3. In line 201, 206, 270 you mentioned aboit good adhesion. What was the method of the testing this adhesion. Did you rely on microscopic examination and lack of defects?

4. How do you explain oxygen presence in the sample showed on figure 8 and EDS results in the table 5?

Best regrads

Reviewer 4 Report

In this work, the wettability of TiZrCuNi filler metal on alumina ceramic and Hastelloy were measured respectively, and the brazing of circular configuration of the Hastelloy C22®/alumina joint was conducted. However, the data is insufficient and the characterization of microstructure is incomplete, so I cannot recommend to publish it in current form. Some suggestions are listed as follow:

1.     The application background of alumina and Hastelloy sealing components in batteries is emphasized in the Introduction. The sealing property and corrosion resistance are two important factors. However, this work does not involve the measurement and analysis of the sealing and corrosion resistance of alumina/Hastelloy components manufactured by brazing.

2.     The microstructure of Hastelloy should be added, and the melting point or DSC/DTA result of TiZrCuNi filler metal should be indicated.

3.     The microstructure of filler metal should be added. The solidus and liquidus temperature or DSC/DTA result of Hastelloy should be indicated, which is very important for the selection of subsequent wetting temperature and brazing temperature.

4.     The shape and size of BTi-5 are very important in the wetting experiment. Because the wetting process can be affected by the contact area between the filler metal and the substrates.

5.     Is Ra or Rms used for surface roughness? It needs to be determined.

6.     As can be seen from Figure 3, there are differences in the holding time of the two groups of wetting systems during the wetting test.

7.     The authors pointed out that there was no typical interface reaction in active braze/ceramic interface, and this conclusion is obtained only by SEM characterization. Obviously, such a conclusion is not convincing. In fact, a large number of literatures have shown that Ti, Cu, and alumina readily generate complex compounds at the interface, such as Ti-Cu-O and Ti-O. It is suggested that further characterization and analysis should be carried out to elucidate the phase composition of the interface, so as to illustrate the wetting spreading mechanism of the filler metal on the surface of alumina ceramics and the formation mechanism of the ceramic brazed joint.

8.     Table 3: The sum of the total composition of each element was not 100%, check it.

9.     For H-C22/BTi-5 brazing interface, it is suggested that further characterization and analysis should be carried out to clarify the phase composition of the interface.

10.  Table 4, table 5: The sum of the total composition of each element was not 100%, check it.

11.  The authors emphasized the great difference between the thermal expansion coefficient of the two base materials, and the brazing of the two materials is successfully achieved through the circular design, while the cracking occurs when brazing the two materials by the plane method. These results are only presented from a macro perspective, without measuring and analyzing the joint stress under different assembly methods. It is suggested to strengthen the research in this aspect from the perspective of simulation or experimental testing.

12.  The performance of brazed joint components will be involved in the service process. In this work, the sealing performance and corrosion resistance are mainly concerned, the results of this aspect should be addressed.

Round 2

Reviewer 2 Report

The revised manuscript can be accepted for publication

Reviewer 4 Report

The revised manuscript can be accepted at its current form.